# Floorball Injuries Presenting to a Swiss Adult Emergency Department: A Retrospective Study (2013–2019)

**DOI:** 10.3390/ijerph18126208

**Published:** 2021-06-08

**Authors:** Stephanie Radtke, Gian-Luca Trepp, Martin Müller, Aristomenis K. Exadaktylos, Jolanta Klukowska-Rötzler

**Affiliations:** Department of Emergency Medicine, University Hospital, 3010 Berne, Switzerland; stephanie.radtke@students.unibe.ch (S.R.); gian-luca.trepp@students.unibe.ch (G.-L.T.); martin.mueller2@insel.ch (M.M.); Aristomenis.Exadaktylos@insel.ch (A.K.E.)

**Keywords:** floorball, Unihockey, injuries, type of injury, mechanism of injury

## Abstract

Background: The popularity of floorball has surged throughout Switzerland in the last 20 years. However, epidemiological studies are still scarce. Objective: To collect information on floorball-related injuries, their severity and approximate cost in adults who presented to our emergency department from 2013–2019. Moreover, to use this information to suggest possible strategies to prevent injuries. Materials and Methods: The study population includes all patients who suffered injuries related to floorball and were then seen at the University Hospital in Bern during a 7-year period. Bern University Hospital, Switzerland, has a comprehensive management system (Ecare) that was used to generate the data for this study. The data were then used to create an injury profile of all cases presented during the said period. Results: A total of 263 injures were recorded from 2013 to 2019. The most common locations for injuries were to the eyes (43.73%), followed by the lower extremities (25.48%). The mean cost per case was CHF 1191.43. However, the vast majority of admissions could be sent home (93.16%) and did not cost more than 500 CHF/case (57.41%). Of the 22 cases that cost more than CHF 2000, 10 were located around the torso and 6 affected one or both eyes. Significant differences were observed between the age groups and treatment area (*p* = 0.008), costs (*p* = 0.008), route of discharge (*p* = 0.023) and type of trauma (*p* = 0.020). Conclusion: Although floorball is still a relatively minor sport, its impact on sport-related injuries must not be underestimated. Injuries to the eyes are particularly common. In our opinion, our findings provide strong evidence that all floorball players (not only children and adolescents) should wear protective eyewear. We conclude that the Swiss Floorball Association (Swiss Unihockey) should mandate the use of protective eyewear.

## 1. Introduction

The sport of floorball is becoming increasingly popular around the world. At the end of 2019, the International Floorball Federation counted around 375,000 registered players in over 70 countries, with 33,500 in Switzerland alone. This excludes the many amateur floorball players, as well as individuals who play floorball in the context of physical education [1]. Although floorball is a very young sport, it was accepted into the ranks of the Olympic Games in 2011 [2,3].

Floorball is a sport that is said to be similar to ice hockey, but without the ice or the skates. Players in teams of six manipulate a light, hollow, plastic ball with graphite sticks. Points are acquired by scoring goals. One match consists of three 20 min sessions with ten-minute breaks in between [1,4,5]. The players must possess good physical strength and motor skills, because of the rapid changes in direction, as well as acceleration, deceleration and feints. Although direct physical contact is strictly forbidden in this sport, accidents can happen. In this context, the floorball players may suffer a variety of different injuries [6,7].

However, in spite of the sport’s susceptibility to injury and its growing popularity, there are not many scientific studies on floorball injuries. A few studies have been conducted in Sweden and Finland on the incidence of injuries in floorball [7,8,9]. Their targeted players ranged from juniors and amateurs to professional national leagues and international championships. Their findings suggest that the players are especially prone to lower limb injuries [7,9,10,11,12,13,14]. For Switzerland, however, there are very few data. Engel and Kälin conducted a retrospective study in 2019 [6]. The goal was to find out if the floorball players of the highest male floorball league in Switzerland could remember suffering from any sport-related injuries during the previous year. Their results also incorporated routine medical examinations that had been conducted up to one year prior to the questionnaires. Their findings suggest that Swiss floorball players suffer from the same injuries as their Northern European counterparts, albeit with a lower incidence. It is striking that Engel and Kälin never mentioned any eye injuries [6], although the risk of eye trauma is recognized [5,12,15,16]. The other study conducted in Switzerland was by Maxén et al., in which they took a closer look at eye and orofacial injuries in floorball and compared their results with Swedish findings [5].

Implementation of protective measures is the responsibility of the federation and its players. Epidemiological studies can, however, be an important tool in identifying the risks of the sport and most common sites of injury and may trigger valuable suggestions for new preventive measurements. For example, Leivo T et al. [15,16] performed two studies on the impact of eye injuries. As a consequence, the regulation was introduced that adolescents must wear protective eyewear when playing floorball, and this in turn has led to a decline in injuries to the eyes. Similar studies have been carried out in almost every type of sport [17].

According to the statistics of the Swiss Council for Accident Prevention (BFU), 11,430 accidents were associated with floorball between 2013 and 2017 [18]. Although this many accidents are associated with floorball alone, wearing protective gear has only become compulsory in Switzerland this season [6]. A number of studies, such as Bro et al. from Sweden in 2017 and Leivo et al. from Finland in 2007, have already shown that protective gear reduces serious injuries and consequently serious complications [12,16].

Increasing healthcare costs are a worldwide problem. The number of visits to emergency rooms is also increasing worldwide, and in addition, ED (Emergency Medicine) care is more expensive than other forms of health service [19,20]. Lamprecht et al. calculated that the costs of physical activity and well-being in Switzerland are a mean of CHF 2100 per case [21]. If the accident costs are put in relation to the number of hours of floorball is played, this results in approximately 100 h of exposure, amounting to about CHF 380 [21]. Moreover, we have also investigated the costs of floorball injuries. This issue has not been examined in previous studies and is relevant to the financial situation of the Swiss Health Care System. These facts show the financial impact on the health care system. It is essential to obtain more detailed information about sport-specific injuries, prevalence and localization in order to reduce costs, floorball injuries and severity of the injury through preventive methods.

In addition to the impact on players’ health, injuries also have socioeconomic impacts. For this reason, floorball injuries are an important area of interest for emergency departments. If there is a fundamental understanding of the casualties, injury types, injury localization and their injury mechanisms, this can provide the basis for adequate patient services and evidence-based strategies for prevention. The epidemiology of related injuries can lead to regulatory changes, equipment changes, training in specific skills and coaching.

This study aims to describe patients with floorball injuries presenting to an adult emergency department between 2013 and 2019 in order to better understand injury patterns and identify risk factors for floorball injury.

The goal of the present study is to investigate floorball accidents in order to record epidemiological data and to analyze the mechanism of injury, its causes and resulting damage. An additional goal is to define simple preventive measurements on the basis of the information obtained.

## 2. Materials and Methods

### 2.1. Study Design

This was a descriptive retrospective study including adult patients (≥16 years) admitted to our emergency department in Bern due to accidents while playing floorball and within the period from 2013 to September 2019. The Department of Emergency Medicine (ED) at Inselspital (University Hospital), Bern, is the only center for major trauma treatment in Bern, with a catchment area of 1.5 million people. In 2019, more than 50,000 patients were treated in the Department 24/7.

### 2.2. Data Collection and Retrospective Analysis

A total of 331 cases were collected in that period, including 263 eligible patients who were further analyzed in our study *(*Figure 1*).* The data for this work were generated from the database of the management system of Bern University Hospital, Switzerland (Ecare, Turnhout, Belgium).

Inclusion criteria were as follows: patient suffered an accident while playing floorball, patient’s age ≥16 years and presenting in our emergency department within the defined period. Swiss medical policy defines adults as patients of 16 years or more; all patients younger than 16 years are treated at the pediatric emergency department.

The following clinical data were extracted from the medical database: diagnosis (type of injury), circumstances of the accident (mechanism of injury), site of injury and accruing costs pertaining to the case (ED visit, hospital stay). The diagnosis was categorized as the major injury with the greatest impact on the patient, and each patient was only allocated to one diagnosis group. If it was not possible to locate one single principle injury, the case was categorised as combined injuries (without life-threatening injuries) or polytrauma. Demographic data such as age and gender were also included, as well as chronological data, such as year of arrival in the emergency department, treatment area, route of admission and route of discharge.

### 2.3. Triage System

At the ED, patients are routinely triaged using an abbreviated version of the Manchester Triage System (Swiss Emergency Triage Scale) [22]. This triage system classifies the priority of treatment for patients into five different triage levels: 1—acutely life-threatening, 2—highly urgent, 3—urgent, 4—less urgent, 5—non-urgent. Specially skilled nurses work on the basis of a defined algorithm for categorization, according to the patient’s reported complaints and treatment priority, using fixed rules that take vital signs into account.

### 2.4. Statistical Analysis

The data were summarized using descriptive statistics (mean values, percentages). The statistical analysis was performed using Stata^®^ 13.1 (StataCorp, The College Station, TX, USA). The distribution of categorical variables is given with the absolute number and the relative number as a percentage. A comparison of the clinical parameters was made between the sex (male/female), year of consultation (2013, 2014, 2015, 2016, 2017, 2018, 2019), and age groups (16–25, 26–35, 36–45, 46–55, 46–55, 56–65, >65 years). A patient with missing data on the treatment performed was excluded from the statistical analysis here.

Categorical variables were analyzed using the chi-square (χ^2^) test and the Fisher exact test. The threshold of significance was set at *p* = 0.05 (two-tailed).

### 2.5. Ethical Considerations

This study was approved by the cantonal (district) ethics committee in Berne (No. 2020-1776) and followed the guidelines of the Declaration of Helsinki and ethical principles for conducting medical research with human subjects [23]. No individual informed consent was obtained. The data were anonymized before analysis.

## 3. Results

### 3.1. Patient Analysis

#### 3.1.1. Annual Distribution

“Floorball” (German “Unihockey”) was identified in both databases for 331 patients. One hundred fifty-three patients (153) were excluded, as their admission was not related to floorball, or the patients were younger than 16.

Twenty-one patients were excluded due to disagreement with the General Consent. For the period from January 2013 to December 2019, two hundred sixty-three (263) cases with floorball accidents were eligible for further analyses *(*Figure 1*)*. Table 1 shows the general patient characteristics. In total, 263 patients with floorball-related incidents were admitted to the emergency department of Bern University Hospital between 2013 and 2019. The annual number of admissions varied between 25 and 58, with the most admissions being in 2018 and the fewest in 2019. There was no significant change in the annual number of patients during the period of 7 years (*p* = 0.722). In the comparison between the analyzed years, no statistical differences were observed for the following parameters: age group (*p* = 0.722); triage group (*p* = 0.927); treatment area (*p* = 0.737); route of admission (*p* = 0.559); route of discharge (*p* = 0.121); type of injury (*p* = 0.704); body part (*p* = 0.257); location of injury (*p* = 0.527); mechanism of injury (*p* = 0.677); costs (*p* = 0.538). The type of trauma was the only parameter that changed significantly over the analyzed years (*p* = 0.006).

#### 3.1.2. Age and Sex Distribution

This retrospective analysis was restricted to patients of at least 16 years, as patients of ages 15 and lower were admitted to and treated by the emergency department of the children’s hospital of Bern. The eldest patient was 80 years of age. The most frequent age group consisted of patients aged 16 to 25 years, with 119 patients (45.25%). Significant differences were observed between the age groups and treatment area (*p* = 0.008), costs (*p* = 0.008), route of discharge (*p* = 0.023) and type of trauma (*p* = 0.020). In the comparison between the age groups, no statistical differences were observed for the following parameters: sex (*p* = 0.546); year of consultation (*p* = 0.927); triage group (*p* = 0.140); route of admission (*p* = 0.663); type of injury (*p* = 0.612); body part (*p* = 0.113); location of injury (*p* = 0.058); mechanism of injury (*p* = 0.570). Two hundred thirty-four patients (88.97%) were male and 29 (11.03%) were female (Table 1). The mean age for women was 27.6 years and for men was 30.9 years. Table 1 and Table 2 list the differences between the genders for all the parameters analyzed. There were no significant differences between males and females for any of the analyzed parameters.

### 3.2. Injury and Clinical Analysis

#### 3.2.1. Location and Type of Injury

The most frequently injured body part was the eye, with 115 (43.73%) admissions, followed by the lower extremities, with 67 (25.48%) cases. The third most common location for injuries was the face and head, with 33 cases (12.55%) (Table 2). The hand was the most frequent location of injury in the upper limbs, with 10 cases (3.8%)

Contusions were the most common mechanism of injury, with 146 (55.13%). In 34 patients (12.93%), the type of injury was a joint distortion, with the ankle region as the most affected body area. Twenty-six (26) patients (9.94%) broke a bone (Table 2, Figure 2), most often in the torso or head. One player injured internal organs, after falling over the low sideboards bordering the playing field. No significant differences were observed between the location and type of injury and sex, year of consultations and age group distribution (Table 1 and Table 2*)*.

#### 3.2.2. Mechanisms of Injury

In 103 patients (39.16%), the reason for an injury was a “blow with a ball”. This was followed by either “tripping over the player’s own feet or their own stick”, with 48 (18.27%) admissions. In a total of 34 (12.90%) cases, the cause for an injury was either “direct player contact”, such as tripping over other player’s feet, or a “direct tackle” from an opposing player. Twenty-five admissions (9.51%) were due to an injury where “no contact” was the mechanism of injury (Table 2, Figure 3). In those cases, the injury was due to the player’s movement, such as an abrupt stop or an abrupt turn. No significant differences were observed between mechanisms of injury and sex, year of consultations or age group distribution (Table 1 and Table 2*)*. The bubble chart (Figure 3) shows the relation between the mechanism of injuries and four main injured body regions. The most common accidents were hitting the eye with a ball (36.1%), followed by tripping over one’s own feet/stick, which caused injuries to the lower extremities (11.0%). Injuries to the trunk and face were caused about equally by all mechanisms but were often associated with direct contact between players (6.1%).

#### 3.2.3. Treatment

The therapy plan depended on the reason for admission. The most common type of treatment was “discharge home with or without analgesia” in 107 cases (40.68%). Fifty-seven (57, 21.67%) patients received a cast or a hard sole. For 4 (1.52%) patients, immediate operation was indicated, for example, surgical treatment of a medial femur fracture. One hundred and forty-six (146, 55.51%) patients were additionally asked to come in again or to visit their family doctor for a follow-up. The treatment area was significantly associated with the age group (*p* = 0.008).

Almost all patients (235, 89.49%) were suffering from a monotrauma. Twenty-two (8.37%) patients suffered non-life-threatening polytraumatic injuries and six (2.28%) patients suffered life-threatening injuries (Table 2).

#### 3.2.4. Admission and Discharge

By far the largest number of patients were self-admitted (*n* = 211, 80.23%). Thirteen (13; 4.94%) patients were brought by an ambulance, and 38 (14.44%) were referrals from other doctors or hospitals. Only one patient was referred to the emergency department by “callmed”, which is a health-insurance-specific medical telephone helpline to evaluate whether a doctor’s referral is indicated or not (Table 1).

Two hundred forty-five (245, 93.16%) patients were able to go home after their emergency room admission. Sixteen (16, 6.08%) were admitted as in-patients. Six patients had injuries to an eye, five to their torso and head, four to their lower extremities and one to their upper extremities. All of those patients were admitted, as the severity of the incident required medical observation or immediate surgery. Two (0.76%) were transferred to a different hospital; one needed surgery and had to be transferred to a hospital closer to home and the other was transferred back to the hospital from where the patient had come.

Furthermore, one patient of age 62 suffered a cardiac arrest whilst playing floorball. The patient passed away in hospital after returning to spontaneous circulation. According to the hospital files, no prior heart conditions were known (Table 1).

The triages were performed at the point of admission by the medical personnel admitting the patient to the ED. A triage is performed to assess how urgently a patient needs to be seen by a doctor for the first time. Triage level 1 (“acute life-threatening”) is the highest level of urgency and a doctor has to see the patient within 10 min of his or her arrival. Triage level 2 means that the patient has the level of “highly urgent” and the doctor must have had his first contact with the patient within 30 min. For triage level 3 or “urgent”, this initial medical contact must be within 90 min. The lowest level of triage is level 4 (“less urgent”) for the ED, for which the patient may have to wait up to 2 h until (s)he sees a doctor. In our study, the patients were mostly triaged on level 3 (193 patients or 73.42%). Fifty (50 patients 19.01%) were triaged on level 2. Only 16 patients (6.08%) were triaged on level 4. Only 4 patients (1.52%) had such a severe injury such that they were triaged on level 1 (Table 1).

Unfortunately, not enough information was available about the circumstances of the individual incidents, such as whether the incident happened during a game or a training session and if the player was wearing any protection.

No significant differences were observed between the route of admission and route of discharge in relation to age groups, sex distribution and year of consultations (Table 1 and Table 2).

#### 3.2.5. Cost Analysis

The range of cost per case was between CHF 93.47 and CHF 26,188.70. The mean cost per case was CHF 1191.43. The inpatient cases all cost more than CHF 2000. Between 3013 and 2019, the admissions associated with floorball had a cumulative cost of CHF 313,346.03.

The year with the highest costs was 2015, with a total of CHF 74,085.22, but the mean cost was CHF 2057.92. It is surprising that 2015 was not the year with the most cases. However, in 2015, three admissions cost over CHF 10,000.00 each. In those three admissions, the patients were treated by the orthopedic team and had to have surgery, for example, due to a fracture. The year with the lowest mean costs was the year with the second most admissions namely 2016 with 51 admissions mean costs of CHF 529.07.

A total of 10 cases made up almost 50% of the hospital expenses, with a mean of CHF 15,478.86 and a total of CHF 154,788.55. Eight of these were admitted to the hospital ward, either for monitoring or surgery. One of the ten patients was admitted to the hospital by the helicopter ambulance and one patient was the patient who suffered a fatal cardiac arrest. There was no medical department that was responsible for all 10 of these patients.

The most expensive department was the orthopedic department, where the mean cost was CHF 8745.93 and the total final costs were CHF 43,729.64 for only five patients. The Ophthalmology Department treated 115 patients, with the second highest hospital expenses of a total of CHF 41,886.58 and a mean of CHF 387.84. Another highly expensive department was the Department of Orofacial Surgery, with only three patients and total costs of almost CHF 40,000.

The age group with the highest total costs was the group of patients between 16 and 25 years of age, with mean hospital expenses of CHF 1176.04 and a total of CHF 139,949.16. The age group with the highest mean cost was the group of patients aged 56–65 years. This was because this group of 10 patients included cases with high hospital expenses of up to CHF 20,000. A statistically significant association was observed between costs and age (*p* = 0.008). Higher costs were generated in the older age groups (56–65 and >65).

## 4. Discussion

### 4.1. Patient Analysis

At the start of our observation period in 2013, members of the official Swiss floorball federation had just hit the 30,000 mark of active players [24]. Floorball gained nationwide popularity during the 1990s and 2000s. Although the sport is still growing, membership has hit a plateau, with 33,976 registered players in the most recent statistics. Our findings reflect these figures, as there is no clear increase in the annual number of injuries. Even though 2018 saw a peak in admissions with 58 cases, the annual case number was mostly around 30, with 2019 even being the year with the fewest records (25 cases).

### 4.2. Age and Sex Distribution

It is not surprising that most of the patients were younger than 35 years, with 45.25% being aged 16 to 25 and 30.04% 26 to 35 years. In general, patients of all age groups were represented, which shows that floorball is no longer a sport only practiced by a few young enthusiasts. The oldest patient was even 80 years of age. Significant differences were observed between the age groups and treatment area, costs, route of discharge and type of trauma.

Unfortunately, no further data concerning the players’ membership status were collected. Because of the game’s simplicity, floorball is very popular outside of non-league settings, such as physical education classes or fun-focused evening games. Almost no statistics are available about those non-league accidents. Such a comparison between the more demanding but better regulated and better-supplied pro-league and the more fun-oriented but less well regulated and trained amateur sports would be of great value. This might help in the development of new strategies to prevent injuries, by taking these two slightly different playing dynamics into consideration.

Of the 263 patients 29 were female. Like men, the majority of injuries were eye-related (51.72% in female), but for female patients, knee injuries were the second most common (13.79%).

Our study found no statistical differences between men and women in any of the parameters. Although we only saw a small number of knee injuries overall, these findings correspond to several other studies about the vulnerability of female floorball players for knee injuries [11].

### 4.3. Injury and Clinical Analysis

The main focus of our study was to portray an adequate picture for acute traumatic floorball injuries. This acute character was mirrored in the number of patients who came 0 to 1 days after the incident, namely 79.08%. On the other hand, not all types of injuries were recorded equally precisely. Mainly chronic injuries, such as overuse, were registered in only small numbers. We recorded five cases of overuse in this time period that led to acute aggravation of the symptoms, whereas studies with different settings or even different definitions of injury have found much more frequent overuse injuries [25,26].

#### 4.3.1. Eyes

The main finding of this study was that the great majority of admissions were caused by an injury to the eye—both in female and male players. Of 263 cases seen during our 8-year observation period, 115 (43.73%) had an eye-related injury. Almost all cases suffered eyeball contusion (240, 91.37%), but at least 76 (28.90) patients were also diagnosed with hyphemia (bleeding of the anterior eye chamber). The most common cause was being directly hit by a ball, and only eight patients were hit by a stick. Hence, 110 cases were monotraumatic and only concerned the eye. Unfortunately, only in 19 cases was it documented whether the player was wearing some sort of eyewear at all, and only five players stated that they had actually worn this eyewear during the accident. Of those five documented cases, one was hit by the ball, but all others were collisions with forces that were too great for the protective eyewear. A specific example of such a case was when the patient was stated in their history to be wearing floorball eye protection glasses, which shattered on impact with the ball.

It may seem that the majority of these eye-related injuries were mild. Ninety-eight (98) cases did not generate more than CHF 500.00 costs, and only five patients had to be hospitalized. The rest could be sent home. However, the gravity of eye injuries must not be underestimated. Most recent sports-related ophthalmological reviews seem to indicate that there is an increased risk after injury of developing future eye-related problems. One study found an increase of 3–4% in the incidence of glaucoma in a 6 month follow-up after an ocular contusion. Another even found an increase of up to 10% in the risk of glaucoma in a 10 year follow up [27].

Our second most common eye-related diagnosis was hyphemia, which also seems to be connected with a lifelong increased risk of glaucoma. In a study by Siu-Chun Ng et al. in 2015, 6.5% of the patients with traumatic microhyphemia were diagnosed with glaucoma 6–98 months after the trauma [28]. Lastly, the eye is a very sensitive organ, and although sometimes no objectifiable cause can be made out, patients may still suffer chronic complaints. This can be seen in a study by Bro et al., where about 20% of the floorball players reported visual impairments such as photosensitivity and reduced night vision even 2–7 years after the incident [12].

Floorball-related eye injuries are a subject of long, ongoing discussions, with the earliest studies dating back as far as 1999 and 1995 [4,29]. Data from those studies claimed that even though floorball back then was only played by a small group of players, approximately 20% of sports-related eye injuries were associated with floorball. Several subsequent studies confirmed that floorball has indeed one of the highest incidences in sports for eye-related accidents [5,15,16].

One follow-up study at the Helsinki University Eye Hospital in Finland compared sports-related ophthalmologic cases before and after the introduction of mandatory protective eyewear in players under 14. It is interesting that they found that, although floorball was still the most frequent sport in Finland to cause eye injuries (accounting for 32% of all cases), the most common age group shifted from the 10–19-year-old patients to the 40–49-year-old patients. Only one floorball-related eye injury under 14 years of age was recorded, which happened during a non-organized practice without eyewear being worn. This shows that, by wearing appropriate eyewear, any sort of floorball-related eye injuries can be prevented [5].

Since then, several other countries have made protective eyewear mandatory for adolescents—most recently Switzerland and Austria. Starting from the 2020/2021 season, every player aged 16 years or younger has to wear protective eyewear [30]. Although in our study no precise statement can be made about eye injuries for adolescents, because this study’s populations only consist of players older than 16 years, our data seem to be consistent with those of previous studies, with eye injuries being the most common in the youngest age group of 16 to 25 (45.25%). Therefore, we certainly approve of the decision for obligatory protective eyewear for the younger players. We even suggest expanding the regulation to all age groups and encouraging the use of protective eyewear in non-organized practice.

#### 4.3.2. Lower Limbs

In total, the lower limbs were injured less often than eyes. In this group, the ankle and knee were the most frequently injured parts of the body: 22/67 (32.84%) and (22/67, 31.34%), respectively.

According to other studies, knee injuries and especially ACL ruptures, are a considerable problem, as they are common in floorball [9]. ACL ruptures are regarded as injuries with severe consequences such as surgery and long absenteeism from play. Because of the emergency setting and the lack of follow-up information in our study, exact diagnoses for ligament tears were difficult to make. Clinical examinations are of poor significance because of generalized pain, and in most cases, no further imaging procedures were carried out, mainly due to the cost. However, we still recorded four cases of suspected or diagnosed ACL ruptures, five ruptures of one of the collateral knee ligaments and at least one meniscal tear. Although only one case resulted in more than CHF 1000 costs, it is important to note that, in many cases, follow-up with possible further imaging was indicated. These costs were not added to the cost in our study, since only emergency costs were included. It is interesting that more injuries involved no-contact than contact with the opponent (75% vs. 25% of all knee injuries). This correlates with other studies, since there seem to be several biomechanical factors influencing the vulnerability to injuries. A series of studies a few years ago showed that, for example, players who move more “stiffly” tend to have more injuries because of the dynamic nature of the sport [31,32,33].

#### 4.3.3. Face and Head

The third most frequently injured body part was the face (12.55% of all cases). Unlike those to the eye, most head accidents were not caused by the ball (only 6.0%) but rather by direct player contact (33.33%) or as a result of a tackle (18.18%).

This seems rather unusual, because most epidemiological studies have found that injuries to the lower extremities, especially the knee and ankle, are the most frequent sites [10,14,34,35]. It can be speculated that this difference was generated because of the slightly different setting in our study. Their data came from mostly prospective studies, following several professional and amateur league teams and recording every injury during training and matches. In contrast, in our study, we only saw injuries deemed severe enough by the player to be presented at our emergency department. It seems that the most common injuries like ankle sprains and strains were treated by the patient himself or by a family doctor. Nonetheless, it is astounding that the percentage of orofacial injuries is so high, especially because it has to be expected that some of those injuries were not fully represented in our study. During our whole observation period, only one tooth fracture was recorded. This seems to suggest that dental injuries were mainly treated by the patient’s dentist or emergency dental service. According to a study interviewing players from Sweden and Switzerland [5], the use of mouth protection in floorball is almost nonexistent, even though the risk for dental fractures was estimated as moderate. It is striking that floorball gives comparable accident numbers to handball—where mouthguards are much more frequent [13,36]. The study concludes that for this frequency of accidents, players should be far better informed as to what to do in case of emergency, since, for example, only 6.5% knew that there was a special protective container for preventing broken teeth.

### 4.4. Type of Injury

In our analysis, the most common type of injury was contusion (55.54%). The next most commonly observed types were distortion (12.93%) and bone fracture (9.89%). As regards the type of injuries, we saw a surprisingly high number of bone fractures, with almost 10% of all recorded injuries being fractures. This is especially striking, since contrary to our expectations, the most frequent fractures were not on the extremities. Fifty percent of all fractures were located in the head or torso, whereas upper and low limbs were each only involved in 25% of the cases. The most common fracture location was the nasal bone (four cases), followed by the ribs and the clavicle (three cases each) (Figure 4). Various authors also analyzed the type of injury suffered by floorball players. According to the study of Löfgren et al. [37], the most common types of injuries were ligament injury (45%), laceration (13%), contusion (13%) and fracture (11%). Another study [35] reported similar results: the most common injury was ankle sprain (45% of all injuries). The results of this study confirm previous findings, even although our results have also reported minor injuries like Achilles tendon rupture, interior organ injury, or muscle overuse.

### 4.5. Mechanisms of Injury

In the current study, 39.16% of the mechanisms of injury were a blow with a ball, followed by “direct player contact”, with 12.92%. Sixty-eight percent (68%) of all injuries are reported to occur without direct contact to the injured body region [29].

In contrast to ice hockey, checking and other harsh body contact are forbidden in floorball. Nevertheless, due to the speed of the game and the tight spaces it is played in, unnatural movements with the potential for injuries happen several times during a game. Normally, these risks are reflected in the number of players injured due to direct or indirect contact with other players. Because of the relatively high numbers of eye injuries compared to the extremities, this was not the case in our study.

### 4.6. Admission, Discharge and Triage

Overall, our study supports the notion that floorball is a relatively safe team sport. Only 13 players had to be brought in by ambulance, whereas four out of five patients came on their own. Seventy-nine percent of the patients were assessed as “urgent” (examination within 30 min needed) or “less urgent” (examination within 90 min needed). Additionally, 19.01% were declared as “highly urgent” (examination within 10 min needed), and only four needed immediate examination. These four cases consisted of one cardiac arrest, two HWS distortion with hypesthesia, and one trauma to the larynx, and all but the cardiac arrest could be sent home. Furthermore, 93.16% of the patients could be discharged home, while 6.08% had to be hospitalized. Four (4) of the hospitalized patients were in need of immediate surgery. Two had one or more fractures of the lower jaw, one had a femoral neck fracture, and one developed erysipelas after a knee wound, which had to be cleaned out.

### 4.7. Cost Analysis

We found that the mean cost of all our cases was CHF 1191.43. This value only incorporates expenses created during the patients’ stay at our university hospital, without follow-up costs. Unsurprisingly, the main cost-generating factor was whether the patient had to be hospitalized, with all 16 hospitalizations generating costs greater than CHF 2000. The most represented type of injury in this category was eye injuries, with 5 of those 22 patients having some kind of eye-related problem, which is of course associated with the sheer number of eye injuries, but nevertheless shows that eye injuries are far from trivial accidents.

### 4.8. Suggestions for Prevention

Our results suggest that there are several possible ways to improve the journey from injury to recovery, which would benefit injured floorball players. We now propose the following suggestions for preventing injury:Protective eyewear in floorball should be mandatory for young adults until at least the age of 16 years. Extending the age limit further and encouraging eyewear in non-organized practise should be seriously considered (Figure 5).As there are numerous injuries to the lower extremities, prevention should be focused on acute ankle and knee injuries. Avramakis et al. suggest that high-shanked shoes with low soles reduce supination in the ankle joint [38].The impact of dental injuries and therefore the knowledge of first aid and the benefit of wearing mouth guards in floorball should be further investigated.More effort is needed in analyzing the biomechanics behind specific movements in floorball, as this leads to a better understanding of the different injury mechanisms.Once this has been achieved, different equipment should be examined to find the optimum compromise between good performance and good protection.The main focus for injury prevention has been laid on competitive sport. As the popularity of floorball grows, further efforts should go into comparing competitive and unorganized play in order to establish whether there are different profiles of injuries and risks and whether preventive measurements in competitive floorball also work in mass sports.

### 4.9. Study Limitations

This retrospective analysis has several limitations. The Ecare health record system only used floorball accidents treated in the emergency department of Berne University Hospital. Thus, floorball injuries treated in other emergency departments or private physicians’ practices were not recorded. Therefore, the number of floorball injuries is of limited general validity, due to the small sample size. The findings of this study may not be representative of floorball-related injuries treated in other types of healthcare facilities or those injuries that received no medical treatment (for example contusion of the ankle). On the contrary, a very large number of treatments for eye injuries can be due to the fact that the eye is a sensitive organ and should be examined immediately by a specialist. Narrative data were used to identify floorball-related accidents. The narrative portion of the Ecare health record system database frequently lacks relevant details of the patient’s history and outcomes (e.g., review of wound after 2 days by a general practitioner), which may lead to errors in selection or interpretation. There are no true exposure data in this study, and this also limits the interpretation of its findings. However, these analyses of people with floorball-related injuries does identify information on diagnosis, lesion site and the mechanism of injury leading to accidents. These deserve further investigation, and this may help to prevent these injuries.

## 5. Conclusions

These analyses have presented a unique insight into Swiss floorball injuries. Strategies to prevent injuries should be developed. The results demonstrate that injuries to the lower extremities and eyes are quite a common problem among floorball players. Since floorball still causes many eye injuries and using protective eyewear reduces the risk of eye trauma, we strongly recommend that floorball players of all age groups wear protective eyewear and suggest that the Swiss floorball association (Swiss Unihockey) mandates its use.

The present analysis is a first step towards identifying risk factors. On the basis of this study, further systematic investigations should be conducted. Including data from the whole country, with the objective of achieving in-depth knowledge of the mechanism and clinical course of floorball-related injuries. To enhance the comparability of the studies and improve classification, standardized assessment tools should be used in the future, and all accidents, including minor incidents without medical care, should be recorded in a register for each team.

Further prospective studies could focus on the monetary impact to the economic system when patients cannot work after floorball accidents, as well as to the healthcare system in general. The financial aspects of this increasingly popular sport would then become clearer.

## Figures and Tables

**Figure 1 ijerph-18-06208-f001:**
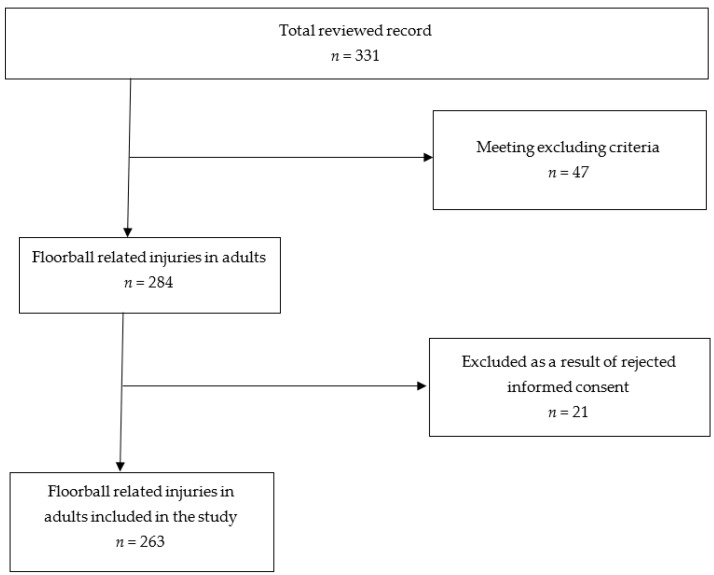
Flow chart of selection of medical records.

**Figure 2 ijerph-18-06208-f002:**
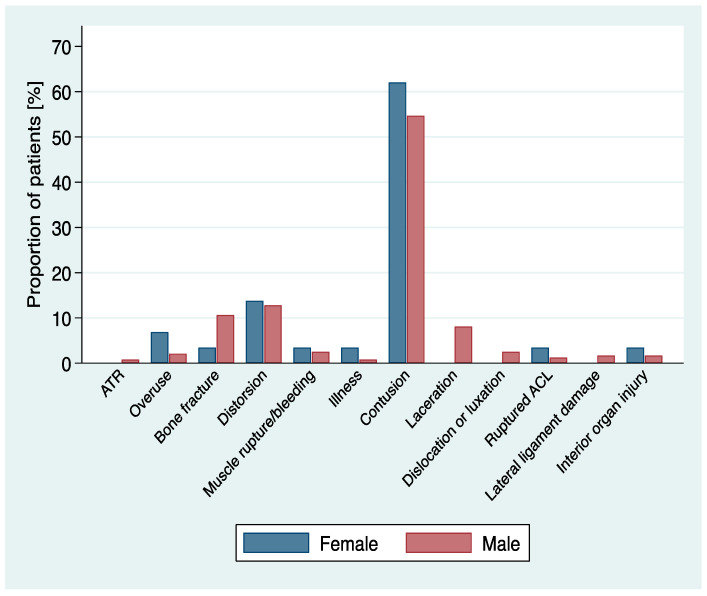
Distribution of type of injury.

**Figure 3 ijerph-18-06208-f003:**
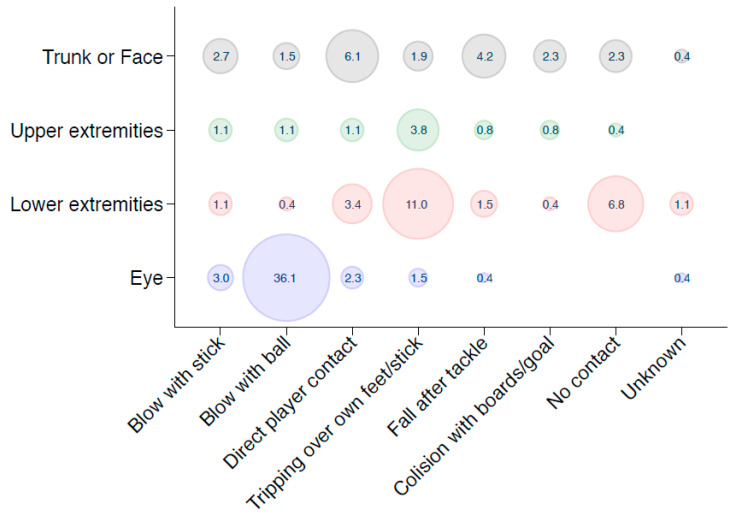
Bubble chart of the mechanism of injuries related to four main injured body regions. In the middle of the bubbles are shown the percentages of all 263 injuries.

**Figure 4 ijerph-18-06208-f004:**
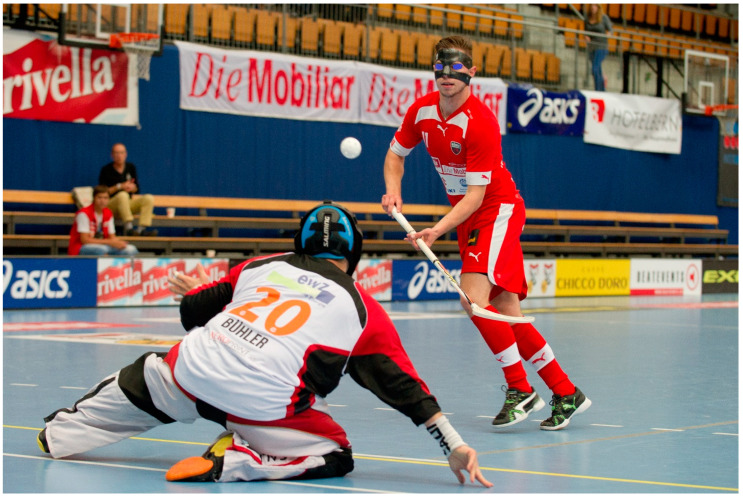
Example of nasal bone fracture mask (Photograph Fabian Trees).

**Figure 5 ijerph-18-06208-f005:**
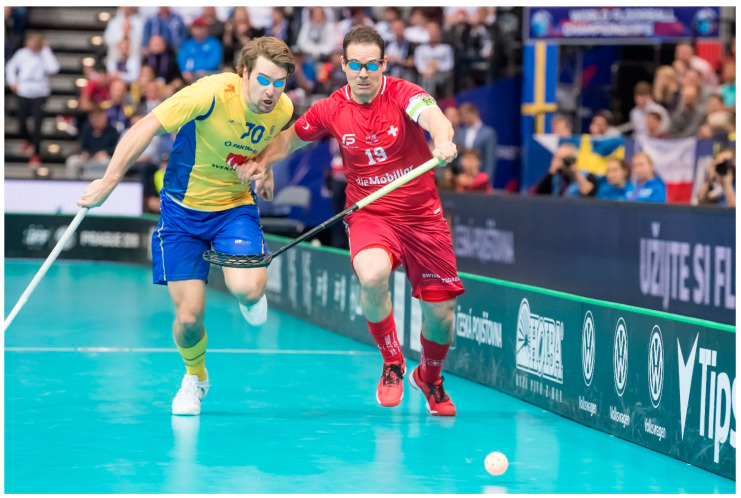
Protective floorball glasses (Photograph Fabian Trees).

**Table 1 ijerph-18-06208-t001:** Characteristics of all patients presenting to the emergency with 263 floorball-related accidents according to sex distribution.

	Total, *n* (%)		Male, *n* (%)		Female, *n* (%)		*p*
**Age group**							0.546
16–25 years	119	−45.2	16	−55.2	103	−44	
26–35 years	79	−30	8	−27.6	71	−30.3	
36–45 years	32	−12.2	3	−10.3	29	−12.4	
46–55 years	19	−7.2	0	0	19	−8.1	
56–65 years	10	−3.8	1	−3.4	9	−3.8	
>65 years	4	−1.5	1	−3.4	3	−1.3	
**Year of consultation**							0.722
2013	28	−10.6	3	−10.3	25	−10.7	
2014	32	−12.2	3	−10.3	29	−12.4	
2015	36	−13.7	1	−3.4	35	−15	
2016	51	−19.4	6	−20.7	45	−19.2	
2017	33	−12.5	5	−17.2	28	−12	
2018	58	−22.1	8	−27.6	50	−21.4	
2019	25	−9.5	3	−10.3	22	−9.4	
**Triage**							0.783
1 (acute life-threatening)	4 (1.5)		0 (0.0)		4 (1.7)		
2 (highly urgent)	50 (19.0)		5 (17.2)		45 (19.2)		
3 (urgent)	193 (73.4)		23 (79.3)		170 (72.6)		
4 (less urgent)	16 (6.1)		1 (3.4)		15 (6.4)		
**Treatment area**							0.964
Fast track	45 (17.1)		5 (17.2)		40	−17.1	
surgery/resuscitation room	83	−31.6	7	−24.1	76	−32.5	
orthopaedic	5	−1.9	1	−3.4	4	−1.7	
cranio-maxillo-facial surgery	3	−1.1	0	0	3	−1.3	
ophthalmology	108	−41.1	14	−48.3	94	−40.2	
neurology	3	−1.1	0	0	3	−1.3	
ears–nose–throat	9	−3.4	1	−3.4	8	−3.4	
internal medicine	6	−2.3	1	−3.4	5	−2.1	
urology	1	−0.4	0	0	1	−0.4	
**Route of admission**							0.424
self-admission	211	−80.2	25 (86.2)		186 (79.5)		
ambulance	13	−4.9	2 (6.9)		11 (4.7)		
family doctor/city emergency practice	13	−4.9	2 (6.9)		11 (4.7)		
other hospital	25	−9.5	0	0	25 (10.7)		
callmed	1	−0.4	0	0	1 (0.4)		
**Route of discharge**							0.532
home	245	−93.2	26	−89.7	219	−93.6	
hospitalized	16	−6.1	3	−10.3	13	−5.6	
transfer to a different hospital	2	−0.8	0	0	2	−0.9	
**Costs**							0.407
<CHF 500	151	−57.4	17	−58.6	134	−57.3	
CHF 501- to 1000	55	−20.9	8	−27.6	47	−20.1	
CHF 1001- to 2000	35	−13.3	1	−3.4	34	−14.5	
CHF 2001- to CHF 5000	12	−4.6	1	−3.4	11	−4.7	
more than CHF 5001	10	−3.8	2	−6.9	8	−3.4	

**Table 2 ijerph-18-06208-t002:** Anatomical distribution of injuries and mechanism of injuries by frequency (*n*) according to sex distribution.

	Total, *n* (%)		Male, *n* (%)		Female, *n* (%)		*p*	*p*
**Type of trauma**							0.826	
Monotrauma	235	−89.4	25	−86.2	210	−89.7		0.56
Combined without life-threatening injuries	22	−8.4	3	−10.3	19	−8.1		0.683
Polytrauma with life-threatening injuries	6	−2.3	1	−3.4	5	−2.1		0.655
**Type of injury**							0.488	
Achilles tendon rupture (ATR)	2	−0.8	0	0	2	−0.9		0.671
Overuse	7	−2.7	2	−6.9	5	−2.1		0.133
Bone fracture	26	−9.9	1	−3.4	25	−10.7		0.218
Distortion	34	−12.9	4	−13.8	30	−12.8		0.883
Muscle rupture/bleeding	7	−2.7	1	−3.4	6	−2.6		0.78
Illness	3	−1.1	1	−3.4	2	−0.9		0.215
Contusion	146	−55.5	18	−62.1	128	−54.7		0.451
Laceration	19	−7.2	0	0	19	−8.1		0.111
Dislocation/luxation	6	−2.3	0	0	6	−2.6		0.383
Ruptured anterior cruciate ligament (ACL)	4	−1.5	1	−3.4	3	−1.3		0.369
Lateral ligament damage	4	−1.5	0	0	4	−1.7		0.478
Interior organ injury	5	−1.9	1	−3.4	4	−1.7		0.518
**Body part**							0.978	
**Lower extremities**								
Heel	1	−0.4	0	0	1	−0.4		0.518
Ankle	22	−8.4	3	−10.3	19	−8.1		0.724
Knee	22	−8.4	4	−13.8	18	−7.7		0.683
Foot, Toe	10	−3.8	1	−3.4	9	−3.8		0.263
Calf	7	−2.7	0	0	7	−3		0.916
Thigh	5	−1.9	1	−3.4	4	−1.7		0.345
**Upper extremities**								
Hand and Finger	10	−3.8	1	−3.4	9	−3.8		0.528
Wrist	4	−1.5	0	0	4	−1.7		0.916
Elbow	2	−0.8	0	0	2	−0.9		0.478
Humerus, Arm	1	−0.4	0	0	1	−0.4		0.724
**Torso**								
Torso, Rib	11	−4.2	1 (3.4)		10	−4.3		0.834
Back, Neck	9	−3.4	0	0	9	−3.8		0.283
Shoulder, Clavicle	8	−3	0	0	8	−3.4		0.312
**Genitals**	2 (0.8)		0 (0.0)		2 (0.9)			0.617
**Face, Head**	33	−12.5	3	−10.3	30	−12.8		0.704
**Tooth**	1	−0.4	0	0	1	−0.4		0.724
**Eye**	115	−43.7	15	−51.7	100	−42.7		0.357
**Location of injury**							0.419	
Eye	115	−43.7	15	−51.7	100	−42.7		0.357
Lower extremities	68	−25.9	9	−31	59	−25.2		0.499
Upper extremities	24	−9.1	1	−3.4	23	−9.8		0.26
Trunk and Face	56	−21.3	4	−13.8	52	−22.2		0.296
**Mechanism of injury**							0.474	
Blow with stick	21	−8	2	−6.9	19	−8.1		0.819
Blow with ball	103	−39.2	16	−55.2	87	−37.2		0.061
Direct player contact	34	−12.9	2	−6.9	32	−13.7		0.305
Tripping over own feet/over stick	48	−18.3	3	−10.3	45	−19.2		0.243
Fall after tackle	18	−6.8	2	−6.9	16	−6.8		0.991
Collision with boards/goal	9	−3.4	0	0	9	−3.8		0.283
No contact	25	−9.5	4	−13.8	21	−9		0.404
No information	5	−1.9	0	0	5	−2.1		0.427

## Data Availability

Data sharing is not applicable to this article.

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
