# Peer review of "Floorball Injuries Presenting to a Swiss Adult Emergency Department: A Retrospective Study (2013–2019)"

_ijerph, 2021, doi:10.3390/ijerph18126208_

Round 1

Reviewer 1 Report

This is a descriptive epidemiological study. Any new or original results are provided.  The conclusions are mostly reccomendations, and are not linked to the statistics, but merely personal opinions. It is an interesting study for administrators, but I don't see where is the science here. 

In addition, statistical methods are not well described.  References are quite few, maybe it can be extended to similar sports ?

The suggested modifications to sport rules (e.g. eyewears) based on traumatology are inappropriate. Every sport has risks and preferred loci of injury, but this not means an athlete must wear every protective garments. This suggestion does not make sense.

I reccomend, in order the paper to be publisheable, to do alternative statistics, for exmple, study the associations between the investigated factors, to build a model for multiple injuries. The results in the present form are of few scientific interest and soundness. A statistical consultant should be appropriate, in order to assess the statistical power of the study. 

Author Response

Bern, 27. Mai 2021

Journal IJERPH

Manuscript ID ijerph- 1236048

Dear Editor

We appreciate your interest and helpful attentative peer-review of this paper:

Floorball injuries presenting to a Swiss Adult Emergency Department. Retrospective study (2013-2019).

We thoroughly considered your comments and these gave us the opportunity to improve and extend the scope of our paper.

Thank you for the both reviewers for critically looking at the study and making suggestions for improvement:

  1. According to the suggestions of both reviewers, we connected with the statistician and performed additional statistical analyses for the results. We estimated statistical differences between genders, years of consultation and age groups for the parameter. In addition, we have visualised on bubble chart relationship of mechanism of injuries related to four main injured body regions.

  1. We have included suggested literature from rewier 2.

  1. We have included (suggestion of revier 2) in the introduction a section with the justification of cost analysis.

Additionally, a quick look is given into the costs of a floorball injury. In the other papers, which are cited here, there was no mentioning of how expensive a floorball injury is. Also considering, that there is an increased interest in the financial situation of the Swiss Health Care System.

  1. In the summary we suggest that above all eye injuries must be recognised and prevention of injuries must be improved by the use of protective glasses. According to the literature, eye-related diagnosis seems to be attached with a lifelong increased risk of glaucoma.

  1. We have also checked for plagiarism and made corrections. Unfortunately, in the part of Material Methods are standard sentences which concern our processes in emergency department, for this reason they are formulations similar to the texts from other our publications.

  1. We also used professional English proofreading services to improve the quality of our publication.

  1. We also send an additional consent form from the photographer.

We certainly hope that our paper is soon to be accepted and published in the International Journal of Environmental Research and Public Health

We wish you best regards and good health.

Jolanta Klukowska-Rötzler

Statement Review

We would like to sincerely thank you for your extensive comments, inputs and questions. The review helped a lot to further improve our paper. We hope that the changes meet your requirements.

Since many comments were related to the comparison of our data with international studies, we have written a general statement below:

In our study, data of patients repatriated to Switzerland from abroad were analyzed. No primary airborne rescue missions were conducted. Repatriations to our center hospital are in most cases planned and prepared, since almost all patients (in our case 91.7%) were previously hospitalized abroad. The repatriations are therefore mostly announced transfers between a foreign hospital and our center hospital. In the studied period, there was no case of primary evacuation in the sense of deployment in a disaster area or accident site. The literature in the field studied was very sparse, with few studies dealing with repatriations from abroad. Many papers - including the literature suggested by the reviewers - refer to primary deployments involving airlift of patients to a hospital for primary care. Here, the suspected diagnoses, severity of illness/injury, etc. are analyzed and compared with existing literature. However, this approach cannot be applied in our case. Due to the previous hospitalization abroad, the severity of illness or injury can no longer be applied to the treatment at our center. For example, results for patients initially suffering from an ischemic cerebrovascular insult could not be compared with existing literature regarding lysis windows and primary care guidelines.

Patients were pretreated for up to 4 weeks, and repatriation was arranged partly at the request of the patients or their relatives (information from the medical history). In this context, our triage system only assesses the situation of the patients after/during repatriation, which can in no way be compared with the initial situation.

Also, the patients' diagnoses were not determined from the foreign reports (which we did not have), but from the outpatient reports from our emergency department, in case of ambulant cases or transfers to other hospitals. Since patients often remained hospitalized with us at the Inselspital after repatriation, we used the discharge diagnoses for evaluation.

Consequently, we found it very difficult to compare our data with pre-existing international literature. Very few studies were identified that did not address the primary care of patients. One study - and additionally the reason for this "follow-up" study - from our emergency department during the study period 2000-2011.

To implement the proposed changes, we again conducted an extensive literature search. Where possible, our statements were compared with and substantiated by international literature.

We have attempted to better ground our data collection, derived results, and discussion points with the information above. Thus, we hope to provide more clarity when reading our study.

Reviewer 2 Report

The authors present a study on injuries in a new sport such as floorball. In this sense, I would like to help that this first article related to these injuries can be published. To do this, I propose to the authors to make some changes:
Introduction
I think it is important in the penultimate paragraph to refer to the importance of knowing these injuries in sport in order to propose new preventive and recovery measures.
Likewise, references from other similar studies from other sports should be included. For example:
Castañeda-Babarro, A., Calleja-González, J., Viribay, A., Fernández-Lázaro, D., León-Guereño, P., & Mielgo-Ayuso, J. (2021). Relationship between training factors and injuries in stand-up paddleboarding athletes. International journal of environmental research and public health, 18 (3), 880.
11430 in letters
Finally, since the authors indicate the cost of these injuries, it would be good if they included a paragraph indicating this aspect.
Material and methods and results
I think that the sample should be worked statistically to determine if any parameter is statistically significant within the different sections (age, sex, year, triage,…). These data could then be deeply debated and the reasons sought. The current way of treating this data is highly descriptive and devoid of interest.
Discussion
The discussion is well focused, but it would need statistical reinforcement to be able to justify the conclusions.
Conclusion
Why the authors indicate that it is not dangerous? What are they based on? Authors must respond to the objective.

Author Response

(The authors gave the same response as above.)

Round 2

Reviewer 1 Report

The paper has been significantly improved.

Now is suitable for the publication.

Author Response

Bern, 01. June 2021

Journal IJERPH

Manuscript ID ijerph- 1236048

Dear Editor

We appreciate your interest and helpful attentative peer-review of this paper:

Floorball injuries presenting to a Swiss Adult Emergency Department. Retrospective study (2013-2019).

We thoroughly considered your comments and these gave us the opportunity to improve and extend the scope of our paper.

  1. Thank you for the second reviewer for critically looking at the study and making suggestions for improvement. We have worked out the following points again with careful precision:
  • Introduction – Costs part, incl. new references.
  • Introduction - Aim of study.
  • Material & Methods – Study design, data collection, tiage system, statistical methods, ethical considerationsincl. new references.
  • Discussion: suggestions for prevention and conclusions. 

  1. We also used professional English proofreading services to improve the quality of our publication.

  1. We also send additional Change of Authorship Form.

We certainly hope that our paper is soon to be accepted and published in the International Journal of Environmental Research and Public Health

We wish you best regards and good health.

Jolanta Klukowska-Rötzler

Reviewer 2 Report

The authors have done a great job in improving the masnucript. however, I believe that the conclusion should be more in line with the proposed objectives.

Author Response

(The authors gave the same response as above.)
